# Inter-Critically Reheated CGHAZ of Ultra-High-Strength Martensitic Steel with Different Cooling Rates

**DOI:** 10.3390/ma16020581

**Published:** 2023-01-06

**Authors:** Wen-Jian Liu, Hong-Ying Li, Wen-Hao Zhou, Deng Luo, Dan Liu, Liang Liang, Ai-Da Xiao

**Affiliations:** 1School of Materials Science and Engineering, Central South University, Changsha 410083, China; 2Xiangtan Iron & Steel Group Co., Ltd., Xiangtan 411101, China; 3Hunan Valin Lianyuan Iron & Steel Co., Ltd., Loudi 417100, China

**Keywords:** ultra-high-strength, martensitic steel, Gleeble thermal simulation, inter-critical reheated CGHAZ, cooling rate, MA constituents, microstructural evolution, toughening mechanism

## Abstract

The mechanical properties of steel’s inter-critically reheated coarse-grained heat-affected zone (ICR CGHAZ) directly affects the service life of machinery equipment. The hardness and toughness of ICR CGHAZ can be optimized simultaneously through tailoring microstructure where cooling rate plays a key role. In this work, the samples with different cooling rates was prepared using thermal simulation. The granite bainite (GB), bainite ferrite (BF) and MA were formed at a 1 °C/s (CR1) cooling rate, while BF and MA were formed at 10 °C/s (CR2) and 30 °C/s (CR3) cooling rates. With the increase of cooling rate, the effective grain size decreased and the number of hard phases increased, resulting in monotonic increase of hardness (260HV3, 298HV3 and 323HV3). CR1 had sparsely distributed coarse slender MA and CR3 possessed tail-head connected MA along PAGBs, which was detrimental to toughness. Therefore, CR2 possessed the best toughness(25J). The microstructural evolution mechanism of ICR CGHAZ with different cooling rates is investigated, corresponding hardening and toughening mechanisms are discussed.

## 1. Introduction

For the purpose of energy conservation and emission reduction, ultra-high-strength steel (UHSS) has become an inevitable choice for machinery equipment such as excavators and cranes [1]. Welding is a necessary processing technology for structural component manufacturing. The welding process largely affects the mechanical properties of the welded joint [2,3,4]. Extensive research has showed that the inter-critically reheated coarse-grained heat-affected zone (ICR CGHAZ) is the location with the poorest fracture toughness in the multi-pass welding heat-affected zone [5,6]. Therefore, methods to optimize mechanical properties, especially toughness, have become popular in the research on high-strength and ultra-high-strength steel [7,8,9].

The chemical composition and welding process of steel determine the performance of the heat-affected zone (HAZ) [10]. The chemical composition of HAZ cannot be regulated by adding welding wire into the welding process, just like with the fusion zone. Therefore, the mechanical properties of the HAZ can only be modified by optimizing the welding process. As an important welding parameter, the cooling rate plays an important role in the microstructure and strength–toughness balance of steel [11]. Guo et al. [12] studied the coarse-grained HAZ (CGHAZ) of UHSS with different cooling rates. The results showed that by adjusting the cooling rate, the strength and toughness of CGHAZ can be comparable to those of base metal. In terms of the ICR CGHAZ, the cooling rate can strongly affect the evolution of MA and grain boundaries in ultra-low carbon steel [13,14]. Huda et al. [15] revealed that the size of MA was finer and the distribution of MA was more benign in 10 °C/s than 2 °C/s in the ICR CGHAZ of X80 pipeline steel, which increased impact toughness. Qi et al. [5] found that the cooling rate during the second thermal cycle can influence the formation of necklace-type MA along PAGBs. Li et al. [16] demonstrated that the low interface energy between matrix and necklace-type MA induced the MA to detach during the impact process, resulting in low-impact energy.

In the actual welding procedure, multi-pass welding is popular due to its welding efficiency. However, research about the effect of the cooling rate on the ICR CGHAZ has mainly focused on ultra-low carbon steel. As for the application of construction machinery steel, the ultra-low carbon steel cannot meet the increasing strength demand. With the increase in carbon content, the microstructure and mechanical properties after dual-pass thermal cycles change obviously. Therefore, research about the effect of the cooling rate on the ICR CGHAZ of ultra-high-strength low-carbon martensitic steel is preferable and indispensable.

In this paper, ICR CGHAZ samples with different cooling rates (1 °C/s, 10 °C/s, 30 °C/s) were obtained using the Gleeble welding thermal simulation. We studied the effect of the cooling rate on microstructural evolution, the hardening mechanism and the toughening mechanism in the ICR CGHAZ of ultra-high-strength low-carbon martensitic steel.

## 2. Materials and Methods

### 2.1. Materials

The experimental steel used in this study was produced via quenching and low temperature tempering, whereby the microstructure of the steel consists of martensite and fine dispersed carbides. The detailed chemical composition (wt.%) was measured using inductively coupled plasma–atomic emission spectroscopy (ICP-AES) and the results are listed in Table 1.

The mechanical properties of ultra-high-strength martensitic steel were measured. Standard tensile test specimens of 25 mm gauge length were prepared according to GB 228.1-2010. The samples for impact tests were machined into standard Charpy V-notch impact specimens of dimensions 10 mm × 10 mm × 55 mm. The impact tests were carried out at −40 °C in accordance with the GB/T 229-2007 standard. The results are shown in Table 2. The continuous cooling transformation curve (CCT) of the steel was investigated in our previous study [17]. From the results, the starting transformation temperature (Ac_1_) and final transformation temperature (Ac_3_) for austenite are determined to be 653 °C and 845 °C, respectively.

### 2.2. Welding Thermal Simulation

The ICR CGHAZ specimens with three different cooling rates were implemented using Gleeble 3500, which were designated as CR1, CR2 and CR3, respectively, for 1 °C/s, 10 °C/s and 30 °C/s cooling rates. The thermal cycle specimens were cut along the rolling direction of experimental steel plates. The specimens were designed as rectangular shapes with dimensions of 86 mm × 11 mm × 3 mm. The real-time temperature of samples was measured by attaching K-type thermocouples which were at the center of the samples. The CGHAZ samples were heated to 1350 °C at 150 °C/s, held for 1 s and cooled to room temperature at 10 °C/s. The ICR CGHAZ samples were first heated to the peak temperature (Tp_1_) of 1350 °C at 150 °C/s, held for 1 s and cooled to 200 °C at 10 °C/s for the first thermal cycle, after which the samples were heated to the second peak temperature (Tp_2_) of 830 °C at 150 °C/s, held for 1 s, and cooled at 1 °C/s, 10 °C/s and 30 °C/s to room temperature for the second thermal cycle. The schematic diagram of the welding thermal cycles of ICR CGHAZ in the process of thermal simulation tests is shown in Figure 1.

### 2.3. Microstructure and Mechanical Properties

After the welding thermal simulation process, the prepared ICR CGHAZ specimens were cut from the middle part of the simulated samples to control the homogeneity. The samples were further machined into sub-size Charpy V-notch impact specimens of dimensions 2.5 mm × 10 mm × 55 mm and were ground before impact tests. The impact tests were carried out three times at each cooling parameter using a JB-S300 drop weight tester at −40 °C in accordance with the GB/T 229-2007 standard. The samples for Vickers micro-hardness were ground and polished before tests, and were measured using a 310HVS-5 hardness tester in accordance with the GB/T 4340.1-2009 standard. Ten micro-hardness values were measured randomly on the surface of each sample at different cooling rates. The average values of toughness and hardness were calculated and compared. The fracture surfaces after impact tests were characterized using scanning electron microscopy (SEM, FEI Quanta-200).

The metallographic specimens were cut from the middle of the samples after thermal simulation tests. Then, the specimens were prepared using the standard mechanical polishing procedures. The samples were then etched with 4% nital solution for 8~12 s to reveal the microstructure. The microstructure of each sample was characterized using scanning electron microscopy (SEM, FEI Quanta-200, Hillsboro, USA). The electron backscattered diffraction (EBSD) specimens were ground and then electrolytically polished at 15 V in 10 vol% perchloric acid alcohol solution. Crystallographic orientation information from samples was characterized using a scanning electron microscope equipped with an electron backscattered diffraction (EBSD, Oxford Nordlysmax2, Oxford, UK) probe. EBSD tests were performed at 20 kV accelerating voltage and 5 μm step size to acquire crystal structure information.

## 3. Results

### 3.1. Microstructure

As shown in Figure 1, the first thermal cycle simulation parameters of ICR CGHAZ specimens were the same as those of CGHAZ. Therefore, the prepared CGHAZ can be approximately considered as the parent microstructure of ICR CGHAZ. Figure 2 shows the CGHAZ microstructure of experimental steel produced via a single-pass thermal simulation. The microstructure was dominated by a lath bainite and martensite (LB/M) matrix separated using prior austenite grain boundaries (PAGBs).

Subsequently, after the double-pass thermal cycle, the microstructure of simulated ICR CGHAZ of ultra-high-strength martensitic steel with different cooling rates is shown in Figure 3. After the second-pass thermal cycle, the microstructure of CR1 was composed of granular bainite (GB), bainitic ferrite (BF) and MA, while the microstructure of CR2 and CR3 consisted of BF and MA. During the second-pass heating process, part of the microstructure has the opportunity to transform into reverted austenite. The PAGB and lath boundaries of bainite and martensite provide nucleation sites for it [18]. As for CR1, the PAGBs observed in the CGHAZ could not be found. When it comes to CR2 and CR3, there were large amounts of MA distributed along the PAGBs. However, the MA of CR2 was discontinuously distributed, while the MA of CR3 was tail–head connected. With the increase in cooling rate, the variety of MA components was distributed from sparse to dense and type from coarse stringer MA (S-MA) to blocky MA (B-MA). The decomposition degree of coarse B-MA gradually increased with the increase in cooling rate.

### 3.2. Electron Backscatter Diffraction (EBSD)

The EBSD results of CR1, CR2 and CR3 are presented in Figure 4. The inverse pole figure (IPF) maps (Figure 4a1–a3) and grain boundary misorientation maps (Figure 4b1–b3) were pictured to analyze the influence of cooling rates on the grain orientation distribution and grain boundary misorientation, respectively. From the inverse pole figure (IPF) maps, different colors in the IPF maps represent different crystal orientations. Additionally, in the grain boundary misorientation maps, the red lines represent high-angle grain boundaries (HAGBs > 15°) while the blue lines represent low-angle grain boundaries (2° < LAGBs < 15°). The change in effective grain size with different cooling rates is shown in Figure 4c. The statistics of high- and low-angle grain boundary fractions are shown in Figure 4d; the distribution of high- and low-angle grain boundaries is shown in Figure 4e.

As shown in Figure 4a,b, effective grain size (with grain boundary misorientation angle higher than 15°) of CR1 can be divided into two types of morphology: equiaxed shape and wide parallel lath shape. Some of the equiaxed shape grains contained a low density of LAGBs while others contained a high density of LAGBs. The wide parallel lath shape grains were separated by HAGBs and there were few LAGBs inside of them. As for CR2 and CR3, there were also two morphologies of effective grain: very fine equiaxed shape and large lath packet shape. The fine equiaxed shape grains contained a high density of LAGBs and the packet shape grains contained some parallel laths. Unlike the wide parallel lath shape grains in CR1, the laths in the packet of CR2 and CR3 were bounded by LAGBs. It is worth noting that CR3 had more fine equiaxed shape grains than CR2. These results were consistent with the microstructure observed by SEM in Figure 3. As shown in Figure 4c,d, from CR1 to CR3, the effective grain size detected using the EBSD method monotonously decreased from 2.23 μm to 1.78 μm. The proportion of HAGBs in CR1, CR2 and CR3 were 30%, 29% and 36%, respectively. Compared with CR2, the HAGBs ratio of CR1 was similar and the effective grain size increased. This means that the HAGBs density of CR2 is higher than that of CR1, CR3 has the highest HAGBs density and CR1 has the lowest. As shown in Figure 4e, the ratio of grain boundaries between 20° and 45° gradually increased from CR1 to CR3.

### 3.3. Mechanical Properties

Figure 5 presents the variation in Vickers micro-hardness and impact energy (−40 °C) of simulated ICR CGHAZ of ultra-high-strength martensitic steel at different cooling rates. The samples of CR1, CR2 and CR3 had Vickers micro-hardness values of 260 HV3, 298 HV3 and 323 HV3, respectively, which monotonously increased with increased cooling rates. However, the impact energy (−40 °C) of simulated ICR CGHAZ was 14 J, 25 J and 20 J, respectively. With the increase in cooling rate, the low-temperature impact toughness of the samples first increased and then decreased, showing a more complex trend. Interestingly, the hardness and toughness of CR2 and CR3 were both better than those of CR1, which was different from the normal law of hardness and toughness, and may be closely related to the microstructure evolution under different cooling rates.

### 3.4. Fracture Characteristics

Figure 6 shows the impact fracture surfaces of ICR CGHAZ of ultra-high-strength martensitic steel at different cooling rates. The brittle fracture features with typical quasi-cleavage planes dominated the fracture surface of CR1. The river patterns can be clearly observed while dimples can hardly be found. There were micro-cracks at the edges of quasi-cleavage planes. As for CR2, the ductile fracture surface with large amounts of dimples was inside the fracture surface. The dimples were fine and equiaxed, evenly distributed on the fracture surface and were formed via a microporous polycondensation mechanism. The fracture surface of CR3 presented the ductile fracture behavior in general, where a large number of dimples existed. However, the toughening characteristics of the fracture surface were weakened, whereby the size and shape of dimples were unevenly distributed. Some dimples were flattened and elongated into shallow and fibrous shapes, showing weakened ductile characteristics.

## 4. Discussion

According to the results, the microstructure of CR1 was quite different from that of CR2 and CR3. Furthermore, as the cooling rate increased, the hardness monotonically increased, while the impact toughness first increased and then decreased. Interestingly, the hardness and toughness of CR2 and CR3 were both better than those of CR1, which were different from the normal laws of hardness and toughness. Therefore, a detailed discussion of CR1, CR2 and CR3 is carried out to investigate the effect of the cooling rate in the secondary thermal cycle on the microstructures and mechanical properties of ICR CGHAZ in experimental steel. Additionally, relative mechanisms of strengthening and toughening are analyzed.

### 4.1. Effect of Cooling Rate on the Microstructure of ICR CGHAZ

The microstructure of the ICR CGHAZ samples consists of re-austenitized and tempered parts due to the heating and cooling of the CGHAZ microstructure in the inter-critical region [19,20]. The microstructure of CGHAZ in this paper was composed of PAGBs and an LB/M matrix. Nakao [21] investigated the peak temperature in the second thermal cycle and proposed MA formation mechanisms when the second thermal cycle peak temperature was between Ac1 and Ac3. The mechanism suggests that the reverted austenite preferentially nucleates and grows up in regions with high dislocations and high-carbon regions such as PAGBs and bainitic lath boundaries. Compared to lath boundaries, PAGBs possess higher dislocation density and carbon content, which are more favorable for the formation of austenite [22].

As for the re-austenitized part of the microstructure, the reverted austenite nucleates in both the PAGB regions and the lath boundaries of the LB/M matrix. The samples of CR1 possessed a very slow cooling rate of 1 °C/s, where the microstructural transformation was significantly different from that of CR2 and CR3. In the PAGB regions, the reverted austenite of CR1 had the full time to nucleate and grow up in an equiaxed shape. The newly formed austenite precipitated ferrite first, and carbon atoms gradually diffused into austenite during the cooling process. Then, coarse ferrite and B-MA were formed and PAGBs were blurred. The enrichment of carbon atoms stabilized the reverted austenite, resulting in a low decomposition degree of MA. The cooling rates of CR2 and CR3 were 10 °C/s and 30 °C/s, respectively. The high cooling rate decreases the bainite transition temperature and increases the driving force. In the PAGB regions, a part of reverted austenite transformed into BF, and the other reverted austenite without phase transformation formed MA. It is generally known that the MA transformation mainly depends on the carbon concentration of reverted austenite and the cooling rate. The diffusion process of carbon atoms is inhibited with the increase in cooling rate in the second thermal cycle. Meanwhile, at the higher cooling rate, the lattice distortion degree of reverted microstructure becomes larger, the microstructure instability decreases and the decomposition degree of formed MA increases. The MA formed is more difficult to grow at higher cooling rates, resulting in more refined B-MA and S-MA. In the PAGB region of CR2, MA components were discontinuously distributed, while in the PAGB region of CR3, the MA components were tail–head connected. This may be because the carbon atoms in CR2 have a longer diffusion time and a higher enrichment degree and the microstructure is more inclined to form fewer and larger MA during the cooling process, thus forming a discontinuous structure. As for CR3, the diffusion time of carbon atoms was extremely short, and the reverted austenite had no time to merge after nucleation on PAGBs, thus forming a head-to-tail structure with a large amount of fine MA. In the LB/M matrix, the formation of reverted austenite of CR1, CR2 and CR3 was all along the lath boundary of bainite or martensite, and the MA generated was mainly S-MA. Due to the decrease in carbon atom diffusion ability, the S-MA was refined as the cooling rate increased.

Due to the second thermal cycle in the inter-critical region, the parts of the LB/M matrix that are not re-austenitized go through a tempering process [23]. The tempered part of CR1 during the second thermal cycle was also quite different from that of CR2 and CR3. As for CR1, the effective grains maintained a parallel lath-shape morphology. Interestingly, the lath bainite and martensite structure usually contain a large number of parallel fine laths, where the lath boundaries consist of LAGBs. However, the lath boundaries in CR1 became wider and consisted of HAGBs. This means that the tempered part of CR1 had gone through a recovery and recrystallization process, where the dislocation density greatly decreased and the substructure transformed into HAGBs. As for the tempered part of CR2 and CR3, the heating and cooling process was so fast that no phase transition occurred in the matrix and the lattice structure remained constant.

From CR1 to CR3, the effective grain size detected by EBSD decreased monotonously. The effective grain size of the re-austenitized part was reduced due to the shortened time for nucleation and growth of austenite at a high temperature. Additionally, the effective grains of the tempered part were refined due to the increased cooling rate that limited the migration of grain boundaries. The microstructure of ICR CGHAZ is composed of a matrix and the inter-critically reheated product, which transforms from the reverted austenite. During the second thermal cycle of CR2 and CR3, there was no phase transition in the matrix, and the corresponding lattice structure remained constant [24]. Previous research has shown that the two intervals of grain boundary distribution of CGHAZ were concentrated at 1–20° and 45–60° because of the K-S relationship [25]. The grain boundary distributed in 20–45° is correlated with the inter-critically reheated product. Due to the ratio of HAGBs in the inter-critically reheated product being higher than that of the matrix, the HAGB ratio increases with the increase in cooling rate. This also explains why CR3 had a higher proportion of HAGBs and a lower effective grain size than CR2. As for CR1, due to the slow cooling rate of the sample, the recovery and recrystallization process of the matrix occurred, which made CR1 unsuitable for the above law. According to the microstructure results, the illustration of the microstructure evolution of ICR CGHAZ with different cooling rates is schematically summarized, as shown in Figure 7.

### 4.2. Effect of Microstructure on the Mechanical Properties of ICR CGHAZ

In this study, the other thermal cycle parameters remained constant for all specimens, and only the cooling rate was altered. As discussed before, the microstructure of CR1 was composed of GB, BF and MA while the microstructure of CR2 and CR3 was a mixture of BF and MA. CR3 had more BF and MA than CR2. As the cooling rate increased, the number of HAGBs in the samples increased. At the same time, the laths of BF in CR1 were wide, and the lath boundaries were HAGBs. The laths of BF in CR2 and CR3 were thinner, and their lath boundaries were LAGBs.

The proportion of hard phases and the density of HAGBs formed in ICR CGHAZ samples were the main factors affecting microhardness [26,27]. On the one hand, the number and area fraction of hard phases were increased from CR1 to CR3. On the other hand, the grain boundaries possessed higher hardness than the interior of grains. From CR1 to CR3, the effective grain size decreased and the HAGB density increased monotonically. Therefore, the microhardness of experimental steel increased with an increasing cooling rate.

In addition, previous research has shown that the toughness of the HAZ is mainly affected by the MA and grain boundaries [28,29]. The stress concentration points such as the MA/ferrite interface can easily become the priority positions for the initiation of microcracks [15]. According to the Griffith theory, the premise of crack nucleation is that the elastic strain energy released is higher than the surface energy of the crack [30,31].
(1)$$\sigma_{c}=\sqrt{\frac{\pi E\gamma_{p}}{\left(1-\nu^{2}\right)D}}$$
where *σ_c_* is the cracking stress, *E* is Young’s modulus, *γ_p_* is the effective surface energy, *ν* is Poisson’s ratio and *D* is the length of the crack. In the process of crack propagation, MA can be seen as the source of cracks. Previous research has calculated the *σ_c_* of MA, finding out that the coarser MA possessed a lower *σ_c_* value and S-MA was more likely to promote cracks to propagate than B-MA. The necklace-type B-MA distributed along PAGBs can result in the enhancement of stress concentration due to strength mismatch in the MA/ferrite interface during deformation. Thus, the necklace-type MA is detrimental to toughness. Meanwhile, some research has demonstrated that HAGBs can effectively arrest cracks and change their propagation [32,33,34]. Because the microstructure of CR1 is obviously different from that of CR2 and CR3, it is discussed separately. Due to the slow cooling rate of CR1, the reverted austenite microstructure went through a long period of growth, and the tempered part of the microstructure underwent recovery and recrystallization. This leads to a reduction in the number of HAGBs, which adversely affects toughness. From CR3 to CR1, the variety of MA types changed from fine island to coarse slender shape and the distribution of MA changed from dense to sparse, which was also unfavorable for toughness. The microstructure of CR1 had a lower critical crack nucleation stress and it was easier for cracks to nucleate. Therefore, the combined effect of MA and HAGBs made the microstructure of CR1 more prone to generating cracks and more difficult to hinder the crack propagation, contributing to the lowest toughness of CR1. As for CR2 and CR3, there was no obvious microstructure transformation in the tempering part during the rapid cooling process. Therefore, the microstructure evolution of reverted austenite becomes the main influencing factor of toughness. The proportion of HAGBs in CR3 was higher because of the higher amount of HAGBs related to the inter-critically reheated product. This is in favor of preventing crack propagation. However, the MA of CR3 was distributed in a tail–head connected network along PAGBs, which had an obvious adverse effect on impact toughness [6,35]. Compared with CR2, although the high HAGBs of CR3 are beneficial to toughness, the morphology and distribution of MA are detrimental to toughness. The adverse effect on toughness is dominant, so the overall toughness of CR3 is lower than that of CR2. According to previous research, fracture morphology is highly correlated with toughness. The fracture surface of CR1 was found to be dominated by quasi-cleavage planes, indicative of brittle fracture mode. Microcracks were also found in CR1, which represented low crack initiation energy. CR1 therefore shows the lowest toughness. The fracture surface of CR2 was dominated by uniform and equiaxed dimples, indicating a ductile fracture mode. Dimples were formed through nucleation, growth and the aggregation of micropores. This process can absorb a large amount of energy, leading to the highest toughness of CR2. The fracture surface of CR3 was also dominated by dimples, showing a ductile fracture mode in general. However, some dimples were flattened and elongated into shallow fiber shapes, showing weakened ductile characteristics. Therefore, the toughness of CR3 is higher than that of CR1 but lower than that of CR2. Based on the above discussion of ICR CGHAZ with different cooling rates, the schematic diagram of the impact fracture mechanism is summarized in Figure 8.

## 5. Conclusions

In the current work, the effect of cooling rate on the microstructure and mechanical properties of ICR CGHAZ of ultra-high-strength martensitic steel has been studied. The major conclusions are as follows:
In the preparation of ICR CGHAZ, cooling at 1 °C/s(CR1) created a microstructure of GB, BF and MA, while cooling at 10 °C/s(CR2) and 30 °C/s(CR3) created a microstructure of BF and MA. From CR1 to CR3, the effective grain size decreased and the number of hard phases increased, resulting in the monotonic increase of hardness. With the increase of cooling rate, the density of HAGBs increased, the MA types changed from coarse slender shape to fine island and the MA distribution changed from sparse to dense, which was beneficial to toughness. In CR3, the formed tail-head connected MA was detrimental to toughness. Therefore, the toughness of samples first increased and then decreased.


## Figures and Tables

**Figure 1 materials-16-00581-f001:**
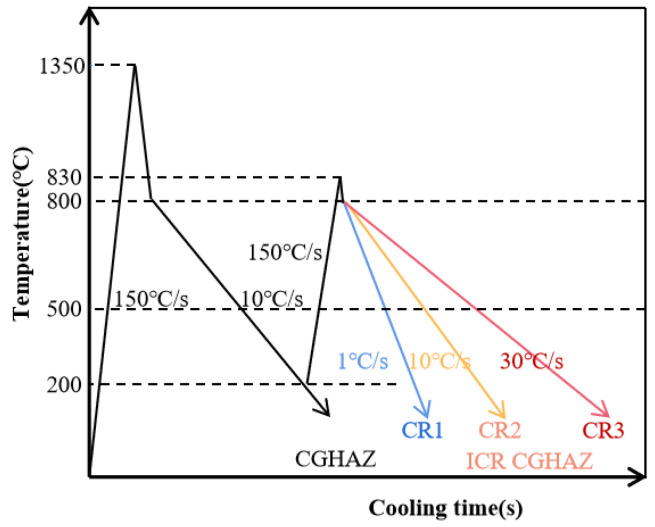
Schematic diagram of welding thermal cycles of ICR CGHAZ.

**Figure 2 materials-16-00581-f002:**
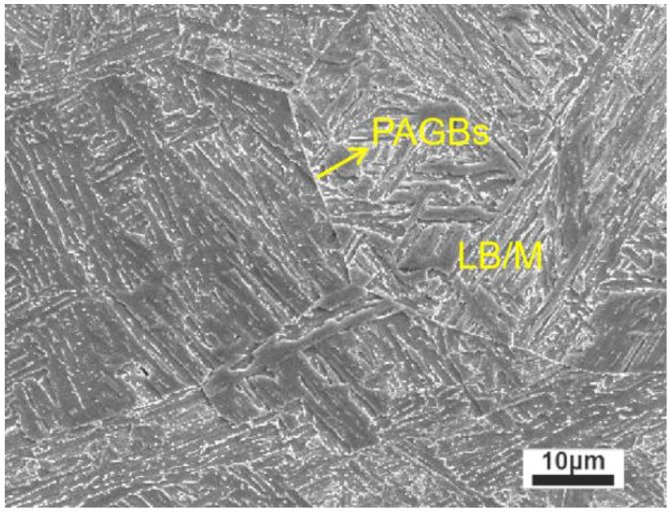
Microstructure of CGHAZ observed by SEM.

**Figure 3 materials-16-00581-f003:**
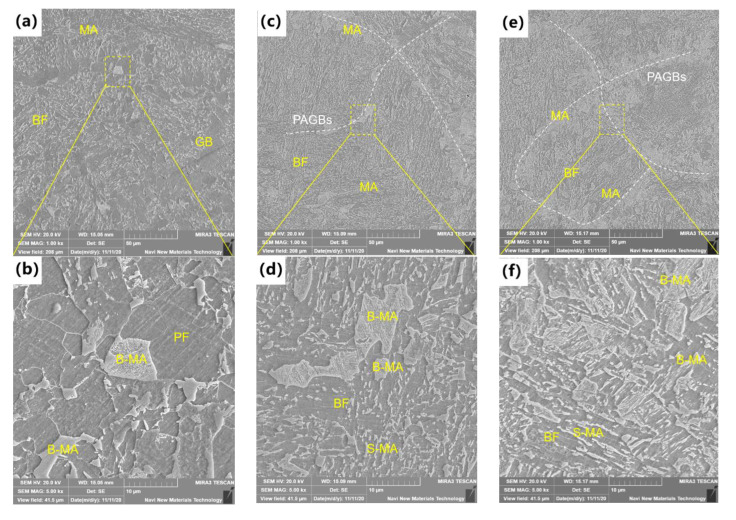
Microstructure of ICR CGHAZ observed by SEM: (**a**,**b**) CR1, (**c**,**d**) CR2, (**e**,**f**) CR3.

**Figure 4 materials-16-00581-f004:**
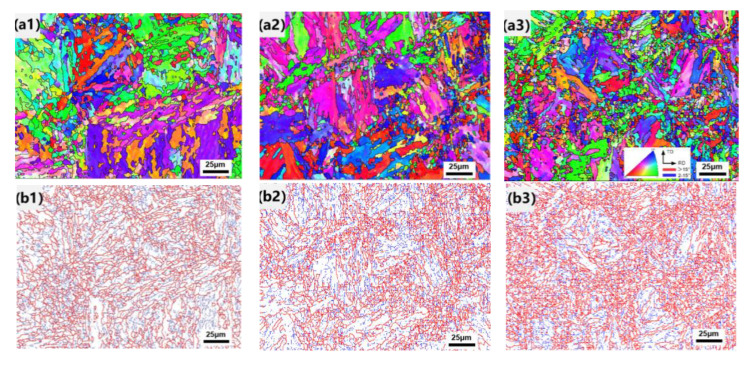
EBSD results of (**a1**–**a3**) IPF maps; (**b1**–**b3**) grain boundary misorientation maps; (**c**) calculated effective grain size graph; (**d**) statistics of high- and low-angle boundaries and (**e**) distribution of grain boundaries.

**Figure 5 materials-16-00581-f005:**
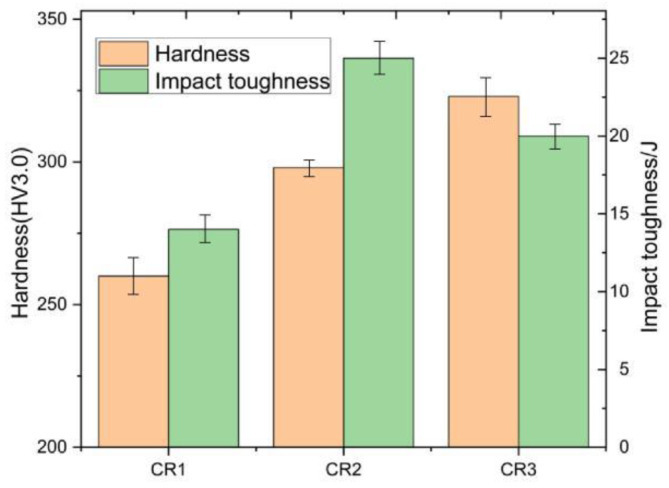
Mechanical properties of ICR CGHAZ under different cooling rates: Vickers micro-hardness and −40 °C impact energy.

**Figure 6 materials-16-00581-f006:**
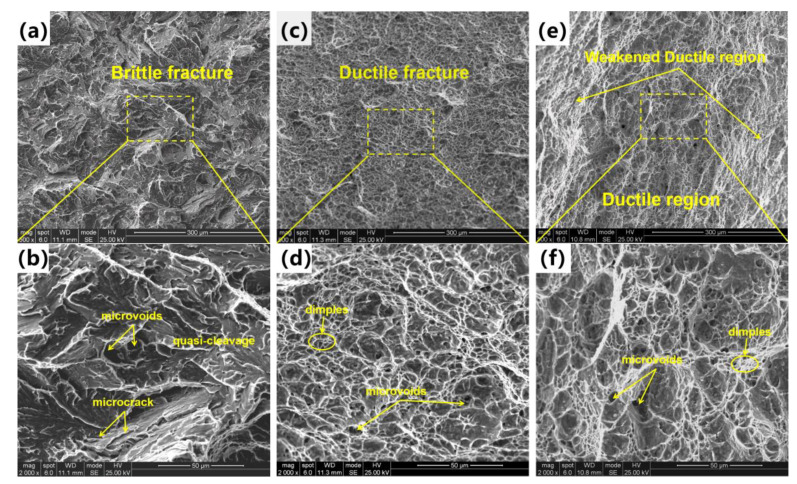
Impact fracture surfaces observed via SEM: (**a**,**b**) CR1, (**c**,**d**) CR2, (**e**,**f**) CR3.

**Figure 7 materials-16-00581-f007:**
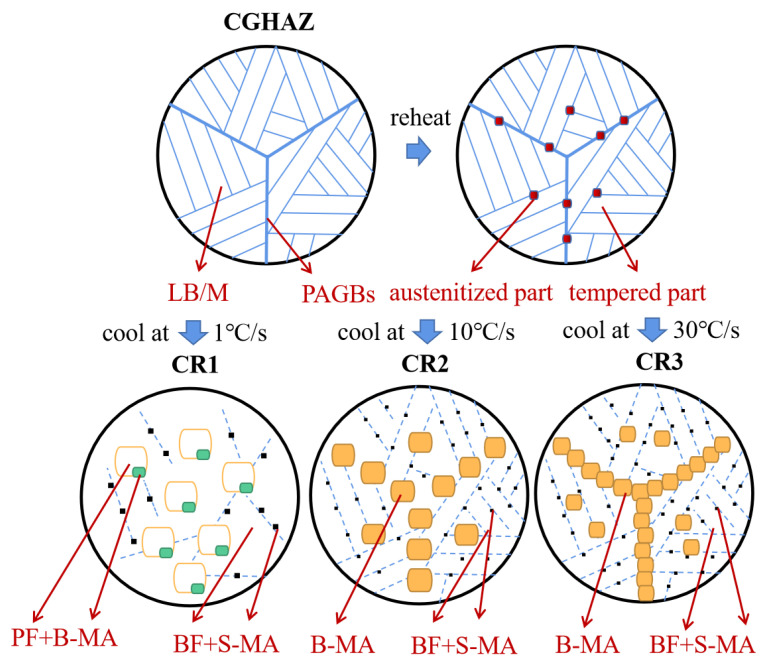
Illustration of microstructure evolution of ICR CGHAZ with different cooling rates.

**Figure 8 materials-16-00581-f008:**
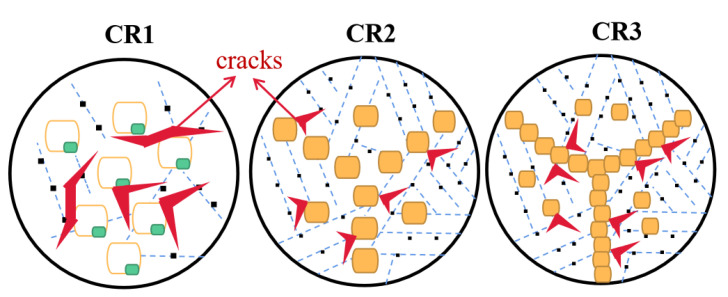
Illustration of impact fracture mechanism of ICR CGHAZ with different cooling rates.

**Table 1 materials-16-00581-t001:** Chemical composition of experimental steel.

C	Si	Mn	Nb	Ti	Cr + Mo + V	B	Al	Ni	P	S
0.15	0.3	1.12	0.022	0.02	0.803	0.0018	0.03	0.32	≤0.009	≤0.009

**Table 2 materials-16-00581-t002:** Mechanical properties of experimental steel.

R_P0.2_/MPa	UTS/MPa	Elongation/%	A_K-40 °C_/J
1260	1388	13	35

## Data Availability

Not applicable.

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
