# Peer review of "Inter-Critically Reheated CGHAZ of Ultra-High-Strength Martensitic Steel with Different Cooling Rates"

_materials, 2023, doi:10.3390/ma16020581_

Round 1

Reviewer 1 Report

Review report: Inter-critically reheated CGHAZ of ultra-high strength martensitic steel with different cooling rates

1.       Shorten the length of the abstract section and add only key information in the abstract section.

2.       Discuss the Novelty and clear application of the work in the abstract as well as in introduction section.

3.       Shorten the length of the introduction section and add key published work and try to make a bridge between current and previously published work: https://doi.org/10.1007/s11665-021-06177-2; https://doi.org/10.1016/j.cirpj.2021.09.002.

4.       How was the composition of the base plate and filler wire selected? How were the mechanical properties of the base metal obtained?

5.       Provide the image of the experimental setup, welded plate and sample prepared from the welded plate.

6.       Discuss the clear detail of the simulation parameters.

7.       Provide the EDS results of the precipitates located in base metal.

8.       Add technical discussion related to metallographic characterization.

9.       Try to relate the variation in mechanical properties with microstructure.

10.    How were the impact results obtained? Add the image of the tested specimen and also discuss the fracture mechanism.

11.    Add detailed discussion about the fracture surface and also mark the brittle and ductile region on the fracture surface: https://doi.org/10.1016/j.engfailanal.2016.06.012.

12.    Overall, work is good and can be accepted after these minor corrections. 

Reviewer 2 Report

The authors have attempted Inter-critically reheated CGHAZ of ultra-high strength martensitic steel with different cooling rates.

The manuscript needs revision based on following comments and suggestions.

1.      Include the quantitative value of key finding in the abstract.

2.      Lack of comprehensive literature support, hence include some more recent relevant literature papers in the introduction sections.

3.      Include high resolution of Fig 4, c,d,e,  and Fig No 5.

4.      Authors should include experimental setup figure.

5.      The manuscript has many typo/grammatical errors. Author need to revise the entire manuscript carefully.

6.      In result and discussion, include the scientific finding and justification with relevant literature.
